# Women’s Depressive Symptoms during the COVID-19 Pandemic: The Role of Pregnancy

**DOI:** 10.3390/ijerph18084298

**Published:** 2021-04-18

**Authors:** Karen Yirmiya, Noa Yakirevich-Amir, Heidi Preis, Amit Lotan, Shir Atzil, Inbal Reuveni

**Affiliations:** 1Department of Psychology, Bar-Ilan University, Ramat Gan 5290002, Israel; Karen.Yirmiya1@post.idc.ac.il; 2Interdisciplinary Center, Baruch Ivcher School of Psychology, Herzlia 4610101, Israel; 3Department of Psychiatry, Hadassah Medical Center, Jerusalem 9103401, Israel; noa269@gmail.com (N.Y.-A.); amit.lotan1@mail.huji.ac.il (A.L.); 4Department of Psychology, Stony Brook University, Stony Brook, NY 11794, USA; heidi.preis@stonybrook.edu; 5Faculty of Medicine, The Hebrew University of Jerusalem, Jerusalem 9112102, Israel; 6Department of Psychology, The Hebrew University of Jerusalem, Jerusalem 9190401, Israel; Shir.Atzil@mail.huji.ac.il

**Keywords:** COVID-19, pregnancy, depression, resilience, experiment

## Abstract

The Coronavirus Disease 2019 (COVID-19) pandemic has multiple ramifications for pregnant women. Untreated depression during pregnancy may have long-term effects on the mother and offspring. Therefore, delineating the effects of pregnancy on the mental health of reproductive-age women is crucial. This study aims to determine the risk for depressive symptoms in pregnant and non-pregnant women during COVID-19, and to identify its bio-psycho-social contributors. A total of 1114 pregnant and 256 non-pregnant women were recruited via social media in May 2020 to complete an online survey that included depression and anxiety questionnaires, as well as demographic, obstetric and COVID-19-related questionnaires. Pregnant women also completed the Pandemic-Related Pregnancy Stress Scale (PREPS). Pregnant women reported fewer depressive symptoms and were less concerned that they had COVID-19 than non-pregnant women. Among pregnant women, risk factors for depression included lower income, fewer children, unemployment, thinking that one has COVID-19, high-risk pregnancy, earlier gestational age, and increased pregnancy-related stress. Protective factors included increased partner support, healthy behaviors, and positive appraisal of the pregnancy. Thus, being pregnant is associated with reduced risk for depressive symptoms during the pandemic. Increased social support, engaging in health behaviors and positive appraisal may enhance resilience. Future studies of pregnant versus non-pregnant women could clarify the role of pregnancy during stressful events, and clarify aspects of susceptibility and resilience during pregnancy.

## 1. Introduction

The Coronavirus Disease 2019 (COVID-19) pandemic created a global health crisis, and its mental health implications are only now beginning to unravel. Studies have reported increased levels of psychological distress, anxiety, depression, and post-traumatic stress disorder (PTSD) in the general population, and specifically in reproductive aged women [1,2,3]. The COVID-19 pandemic has multiple ramifications for pregnant women, including stress surrounding the uncertainty related to prenatal care and the risk of prenatal exposure to the virus [4]. 

The psychological impact of major stressful events on pregnant and postpartum women has been previously studied with conflicting results [5]. Some studies suggest that peripartum women are at high risk for developing mental health problems in the wake of major stressors, with high rates of depression and other perinatal mental health problems [6,7,8,9]. Contrarily, other studies demonstrate resilience among pregnant and postpartum women during natural disasters [10], as well as lower distress levels in the long term compared to women who were not pregnant when the disaster occurred [11]. During the COVID-19 pandemic, several studies revealed that pregnant women have higher rates of anxiety and depressive symptoms when compared to women who were pregnant before the pandemic emerged [12,13,14,15,16]. These studies mostly compared women who were pregnant during the COVID-19 pandemic to women who were pregnant before the pandemic. Interestingly, a study from China, in which pregnant and non-pregnant women were compared during the COVID-19 pandemic, yielded reduced risk for depression, anxiety, insomnia and PTSD among women who were pregnant [17]. A more recent study that compared postpartum women to women who were never pregnant did not show any significant differences in general anxiety and depressive symptoms [18]. There is no doubt that the pandemic affected women’s lives intensely; however, there may also be unexpected positive aspects [19]. Hardships experienced during the perinatal period have been previously shown to yield positive outcomes, such as post-traumatic growth [20]. Further studies are warranted to delineate the role of pregnancy on the susceptibility to stress among reproductive-age women. 

It is unclear whether pregnancy is a time of increased vulnerability for the onset or worsening of a mental illness, as some research suggests [21], or rather it is more specific for the postpartum period or for a subgroup of vulnerable women [22]. Research suggests that for some women pregnancy serves as a protective developmental stage, as suggested by the low suicide rate during pregnancy and during the two years after giving birth [23,24]. Moreover, attenuated emotional and physiological responses to stress during pregnancy may also contribute to enhanced resilience in pregnant women [25,26]. This may be related to the effect of estrogen on cortisol, as evidenced by blunted cortisol responses to stress in postmenopausal women receiving estrogen treatment [27,28]. During pregnancy, women cope with stress in various ways. Dispositional optimism, social support, and physical activity are related to better psychological and physical well-being during pregnancy [29,30]. A longitudinal study from China revealed that education levels and resilience were associated with both adaptive and maladaptive coping, and that women with maladaptive coping styles experienced more postpartum depression [31]. Among a Spanish cohort of pregnant women, lower resilience was related to obsessive and catastrophic thoughts about the pandemic, while adaptive coping strategies, such as physical exercise or relaxation, were effective in coping with pandemic-related restrictions [32]. Partner support, emotional support and being outdoors have been found to increase resilience and positive coping during the pandemic for women in the perinatal period [33,34]. Determining pregnancy-specific traits, resources, and behaviors that improve pregnant women’s mental health during crises could promote resilience during pregnancy. 

Untreated depression during pregnancy is associated with detrimental long-term effects on the mother and the infant [35,36,37,38]. Women who experience stress during pregnancy, even without depression, are less likely to maintain optimal health behaviors, including healthy eating, vitamin use, and exercise [39,40]. Therefore, examining the potential effects of the pandemic on the mental health of pregnant women is crucial for early identification of depressive symptoms during pregnancy, and the prevention of the long-term outcomes on both the mother and her offspring. 

The present study is part of an ongoing international collaboration aiming to investigate the psychological implications of the COVID-19 pandemic among pregnant women [4,34,41]. The results from the United States (US) sample revealed that pregnant women experienced substantial anxiety during the COVID-19 pandemic [4,34]. Increased stress related to the pandemic, as measured by the Pandemic-Related Pregnancy Stress Scale (PREPS), was associated with an increased risk for anxiety above and beyond sociodemographic and obstetric variables [4,42]. The aim of the current study was to determine the risk for depressive symptoms in pregnant and non-pregnant women in Israel during the COVID-19 pandemic, and to identify bio-psycho-social contributors that increase the risk of experiencing depressive symptoms. The results of this study could contribute to the understanding of the role of pregnancy on the vulnerability for depression during times of global crisis.

## 2. Materials and Methods

### 2.1. Participants and Study Design

During the second week of May 2020, we recruited a sample of 1380 women through women-related and pregnancy-related social media groups and public pages on Facebook. The advertisements invited non-pregnant women to share their experiences during the COVID-19 pandemic and pregnant women to share their pregnancy experiences during the COVID-19 pandemic. Our inclusion criteria were age 18 years or older and being able to read and write in Hebrew. There were no other exclusion criteria. Women who wished to participate in the study signed an electronic informed consent, and completed the questionnaires online through Qualtrics, a secure online survey system. After identifying 13 duplicate responses, the final sample included 1114 pregnant women: 81 in the first trimester; 430 in the second trimester; 570 in the third trimester; and 33 women who did not report their gestational age but were identified as being pregnant, and 256 non-pregnant women. The majority of women were born in Israel (N = 1187, 86.6%). Participants who completed the questionnaire were enrolled in a raffle to win a ILS 600 gift card (equivalent to approximately USD 150). The study was approved by the Hebrew University of Jerusalem Institutional Review Board on 11 May 2020 (approval number 114120).

### 2.2. Instruments

Sociodemographic variables included age, years of education, economic status (below average/average or above average), relationship status (married or cohabiting/some or no relationship), current employment status (working or studying/homemaker, unemployed or receiving disability benefits), and number of children under 18.

COVID-19-related variables included having direct contact with an individual medically diagnosed with COVID-19 (no/yes), thinking that one had COVID-19 even if one was not tested (no/unsure/yes), and having access to outdoor space (yes, whenever I want/sometimes/rarely).

Depressive symptoms were assessed using the Patient Health Questionnaire-2 (PHQ-2) [43], which includes the first two items of the PHQ-9 [44]. The two questions inquire about the frequency of depressed mood and anhedonia over the past 2 weeks, on a scale of 0 (not at all) to 3 (nearly every day). The final scores range from 0 to 6 with a higher score indicating a greater risk for depressive disorder. In addition, a cut-off of ≥3 was used. The PHQ-2 has been shown to be efficient to rule out depression in 60–80% of pregnant women [45]. In pregnancy, the sensitivity of the PHQ-2 was 69–84% and the specificity was 79–84% [45].

Stress related to pregnancy during the COVID-19 pandemic was assessed using the PREPS questionnaire, a novel instrument created by Preis, Brittain, and Lobel [42], a multidisciplinary research and clinical team with expertise in developing validated instruments to assess prenatal maternal stress and coping. Item themes were based on news articles and media interviews regarding women’s experiences during the COVID-19 pandemic, as there was limited research available at the time [42]. Item wording was tested for face validity by pregnant and non-pregnant women before the Stony Brook COVID-19 Pregnancy Experiences (SB-COPE) Study launch. The PREPS questionnaire was translated to several languages, and was found to have good psychometric properties in different populations [46,47]. For the current study, the scale was translated to Hebrew using the forward-and-back translation technique by bilingual researchers (see Appendix A
Appendix A). The instrument includes 15 items describing thoughts and concerns that pregnant women might have owing to the COVID-19 pandemic, rated on a scale from 1 (very little) to 5 (very much). The PREPS includes three independent, internally consistent, factors: stress associated with preparations for birth and the postpartum period (7 items; PREPS-Preparedness), stress associated with worries about perinatal COVID-19 infection (5 items; PREPS-Infection), and positive aspects of the pandemic in the context of pregnancy (3 items; PREPS-Positive Appraisal) [42,48]. Confirmatory factor analysis (CFA) was conducted on the Hebrew version to replicate the three-factor structure of the PREPS previously identified, which yielded a good model fit (CFI = 0.929, TLI = 0.910, RMSEA = 0.078). The internal consistency of the subscales PREPS-Infection and PREPS-Preparedness was relatively high (α > 0.8). Although the internal consistency of the PREPS-Positive Appraisal subscale was lower than the α = 0.70 criterion, inter-item correlation coefficients of all items were > 0.32. Scale scores were calculated as the mean response to items on the corresponding factor.

General stress was assessed using a single general question: “How much stress do you currently have in your life?”. Participants were asked to score their answer on a 5-point scale ranging from 1 (very little) to 5 (very much).

Anxiety symptoms were assessed using the Generalized Anxiety Disorder-7 (GAD-7) scale [49], a 7-item self-report instrument to assess anxiety symptoms. Respondents report the frequency of symptoms over the last 2 weeks on a scale ranging from 0 (not at all) to 3 (nearly every day). Individual scores were calculated as a sum of item responses ranging from 0 to 21. The GAD-7 was found to have good reliability and validity for screening anxiety among pregnant women [50,51].

Social support included two single-item questions on perceived support from family or friends and perceived support from a partner (1 = very little to 5 = very much).

Health behaviors were assessed by a single-item question assessing the extent to which women were practicing health behaviors, such as taking vitamins, exercising, and sleeping enough (1 = very little to 5 = very much).

Obstetric factors included gestational age (in weeks), self-reported high-risk pregnancy status (no/yes/unsure), and whether participants had a prenatal care appointment canceled or rescheduled owing to the COVID-19 pandemic (no/yes).

### 2.3. Statistical Analysis

Statistical analysis was conducted using SPSS for Windows, version 23 [52] and AMOS 21 [53]. We used χ^2^ and *t* tests to examine differences between pregnant and non-pregnant women, with *p* < 0.05 considered statistically significant. Propensity-score matching was performed using Stata version 16.1 [54], KMATCH module [55] with the default parameters, which matches treated and untreated observations with respect to covariates based on propensity-scores, including estimation of treatment effects based on the matched/balanced observations, while including post-matching regression adjustment. For post-estimation evaluation of data balancing, kernel density estimates before and after matching were generated using the “kmatch density” command. Pearson’s correlation coefficients were used to assess associations among all study variables. A hierarchical linear regression analysis was conducted among the pregnant women’s cohort to predict the risk for depressive symptoms from all the sociodemographic, psychological, and medical factors. A second hierarchical linear regression analysis was conducted among the pregnant and non-pregnant cohorts with the same predictors as the first regression, to examine the contribution of these variables above and beyond the factor of pregnancy in predicting depressive symptoms. 

## 3. Results

### 3.1. Group Differences between Pregnant and Non-Pregnant Women 

Pregnant women reported significantly fewer depressive symptoms than non-pregnant women (see Figure 1a, Table 1). The difference in the depressive symptoms score remained significant even after controlling for age, education, relationship status, income, number of children and employment status (*F*_(1)_ = 8.48, *p* < 0.005). Our data demonstrate that 15.6% of pregnant women were depressed, as defined by a clinical cut-off of 3 or higher [56], whereas 21.9% of non-pregnant women reached this criteria (*X*^2^_(1)_ = 5.83, *p* = 0.02). No significant differences were observed between pregnant and non-pregnant women in levels of stress and anxiety (Table 1). Pregnant women were less likely to think that they had COVID-19 than non-pregnant women (Figure 1b, Table 1). Pregnant women also reported having more social support and engaging in more health behaviors than non-pregnant women (Table 1). All differences remained significant following the Bonferroni correction for multiple comparisons, except for education level.

To account for potential sample selection bias, we applied regression-adjusted propensity-score matching using a combination of age and education (Appendix A) or age, education, income and employment status (Appendix A) as covariates. The models showed that while providing the expected superior propensity-score matching, significant effects of pregnancy on PHQ2 scores were consistent with the non-matched results.

### 3.2. Multivariate Correlates of Depressive Symptoms among Pregnant and Non-Pregnant Women

Being pregnant significantly predicted lower levels of depressive symptoms. Furthermore, women who had lower income, fewer children, women who were unemployed, and those who thought that they had COVID-19 were found to be at higher risk for depressive symptoms. Women with higher perceived partner support and women engaged in more health behaviors had fewer depressive symptoms. The current model explained 9% of the variance in depressive symptoms. The results of the hierarchical linear regression among pregnant and non-pregnant women, with the depressive symptoms score as the dependent variable, are shown in Table 2. The associations between all predicting variables are detailed in Appendix A.

### 3.3. Multivariate Correlates of Depressive Symptoms among Pregnant Women 

Among the sociodemographic variables tested, pregnant women who had lower income, fewer children, and women who were unemployed reported elevated depressive symptoms. Women who believed that they had COVID-19 (even if not tested) also reported higher depressive symptoms scores. Women with higher perceived partner support and women who engaged in more healthy behaviors reported fewer depressive symptoms. Among the obstetric variables, having a high-risk pregnancy and earlier gestational age significantly predicted higher depressive symptoms scores. The above-mentioned variables together explained 12.5% of the variance in depressive symptoms among pregnant women. Finally, the PREPS-Preparedness stress scale predicted higher depressive symptoms scores, the PREPS-Positive Appraisal scale predicted lower depressive symptoms scores, while the PREPS Infection stress scale did not predict depressive symptoms. The final model explained 18.5% of the variance in depressive symptoms. The results of the hierarchical linear regression among pregnant women, with the depressive symptoms score as the dependent variable, are shown in Table 3. Pearson’s correlation coefficients among all predictive variables are presented in Appendix A.

## 4. Discussion

The outbreak of COVID-19 in Israel began at the end of February 2020 and reached the first peak during March and April 2020. During this time, an almost complete lockdown took place for 8 weeks, with substantial social and economic restrictions continuing for months. Furthermore, information on the effects of the virus on pregnancy outcomes and fetal development was scarce. Fear from being infected with the virus, social isolation and the economic turmoil caused by the pandemic had a notable impact on the mental health and wellbeing of the general population, with increased risk for depression and anxiety among women [1,57]. The results of the present study indicate that during the first wave of the COVID-19 pandemic, pregnant women had fewer depressive symptoms than non-pregnant women. Moreover, pregnant women were less likely to think that they have COVID-19 than non-pregnant women. Our results are in line with previous studies that compared pregnant to non-pregnant women’s responses to the COVID-19 pandemic [17,58]; however, these studies were with relatively smaller samples. Our results may indicate that being pregnant contributes to enhanced resilience in times of crisis and severe stress, and warrant further research on the underlying mechanisms. 

Psychosocial determinants are well-known factors that affect vulnerability to experience depression throughout life, and specifically in the perinatal period. In the present study, pregnant women reported experiencing more perceived partner support than non-pregnant women, and partner support significantly predicted lower levels of depressive symptoms among pregnant and non-pregnant women during the pandemic. Partner support is a well-known protective factor against depression during the perinatal period [59,60,61,62], especially in times of crisis and stress [63]. These results are consistent with reports of pregnant women during the COVID-19 pandemic that demonstrate higher perceived social support was associated with less depressive symptoms [21], whereas a lack of social support was related to increased risk for perinatal depressive symptoms [64]. In a mixed-methods pilot study from the US, pregnant women who reported feelings of isolation and loneliness also reported that partner support was the most important protective factor which helped them to cope with the pandemic [33]. Moreover, we found that health-related behaviors were significantly more frequent among pregnant women than among non-pregnant women, and that these behaviors also significantly predicted fewer depressive symptoms. The relationship between health behaviors and depression is well-established in the literature [65,66,67], and specifically during pregnancy [68]. This relationship has been reported both at non-challenging times and during the pandemic [21,69]. In a parallel study in the US, pregnant women who were engaged in more positive health behaviors were also found to have less anxiety symptoms [34]. A Finnish study monitoring pregnant women’s daily patterns of heart rate variability, physical activity and sleep data showed that these variables were associated with coping better with pandemic-related restrictions [70]. Therefore, improving women’s social support networks and encouraging socially distant healthy activities, such as online fitness classes suitable for pregnant women, may be targets for enhancing resilience to depression among pregnant women during the pandemic.

We found that vulnerability for depression is also associated with financial and employment problems. In Israel, pregnant women are protected by law from layoffs during pregnancy and the postpartum period [71], and women receive paid maternal leave for the first fourteen weeks after birth [72]. These factors may protect against experiencing depressive symptoms and enhance resilience among pregnant women, especially in light of the detrimental economic and psychosocial effects of the COVID-19 pandemic. This is supported by recent findings from Canada, which showed that pregnant women with high income compared to low income were less likely to experience distress and psychiatric symptoms during the COVID-19 pandemic [12]. Another recent study from the US revealed that COVID-19-related financial stress was associated with a significantly increased likelihood of depression during pregnancy [73]. Implementing policies related to job security and stable income for pregnant women may serve as protective factors and improve their mental health. 

Among the obstetric variables measured in the study, depressive symptoms were inversely correlated with gestational age and were positively correlated with a high-risk pregnancy. Pregnancy complications are well-known risk factors for postpartum and antenatal depression [74,75]. Women who have a high-risk pregnancy should be considered at risk for depression, especially during the COVID-19 pandemic. Our findings, which show increased levels of depression earlier in pregnancy, are not in line with research reporting higher levels of depression towards the end of the pregnancy [76]. Interestingly, a recent longitudinal study of 135 pregnant women showed an increase in depressive symptoms after COVID-19 compared to measurements before the pandemic started. However, the levels of depressive symptoms after COVID-19 were as high as those during early pregnancy [77]. It is unclear whether depression levels earlier in pregnancy in our cohort were related to the unique circumstances of the initial stages of the pandemic, or rather reflect a possible rise in depression symptoms during the beginning of pregnancy. Further research is needed to confirm and understand the results regarding the relation between gestational age and depressive symptoms. 

Depression scores among pregnant women were predicted by several factors related to the pandemic. First, women believing they have COVID-19 (even without being diagnosed) was associated with a higher depressive symptoms score. Second, the Preparedness Stress score predicted a greater likelihood of depressive symptoms, indicating that women who were worried about being unprepared for the birth due to the pandemic were more likely to experience depressive symptoms. In line with these results, a recent study of women who were pregnant during the COVID-19 pandemic found that higher depressive symptoms were associated with more concerns about not getting the necessary prenatal care [21]. We also found that the pandemic-related positive appraisal score reflected resilience, as it was inversely associated with the depressive symptoms score. The transition to motherhood is an opportunity for personal growth and finding meaning in life and gratitude [78]. Positive appraisal, in other words, focusing on the “positive aspects of a stressful situation when one possesses the intrapersonal and tangible resources that help to ensure a favorable outcome” [79], has been shown to be predictive of lower distress levels in pregnant women [79,80]. Having a positive attitude towards the effects of the pandemic on a woman’s pregnancy and expected birth may represent a coping mechanism, which allows women to cope and adjust to the major life change they are about to undergo. Implementing psychotherapeutic interventions that enhance the use of positive appraisal and changes in maladaptive cognitive processes may contribute to reducing perinatal depressive symptoms during the pandemic.

There are several limitations to this study. First, the study was conducted online with self-report questionnaires. However, this enabled the recruitment of a large number of women during a short period of time, which was crucial in light of the rapidly progressing situation worldwide. Second, it should be noted that this is a cross-sectional study, which provides a snapshot of the mental health of women in May 2020. During that time, there was a relatively small number of COVID-19-related infections and death cases (an average of 2.188 new infections and 0.233 new death cases per million people per day [81]). However, the lockdown and movement restrictions applied, as well as the uncertainty regarding the influence of the COVID-19 pandemic worldwide, created an atmosphere of stress and anxiety. The results may not necessarily signify a direct causal association between pregnancy and resilience, but rather suggest that other factors that are related to pregnancy, such as social connection and support, may mediate the relationship between pregnancy and resilience. Future stages of the current study and other studies will be able to evaluate the course of mental health symptoms and their sequela throughout the pandemic. Third, this type of self-selected sampling could be biased because most participants were born in Israel, married, and secular, limiting the generalizability of our results. The pandemic has disproportionately impacted racial minorities and lower-income families [82], which are underrepresented in our cohort. Moreover, more severely depressed women are probably less likely to go online and be willing to participate in a study. Lastly, PHQ-2 may not be ideal for the assessment of depression during pregnancy, as there are different instruments that specifically address anxiety and depression during the perinatal period. However, this instrument was chosen as it is a concise and internationally validated screening tool for depression [43,56,83,84]. The psychometric properties are good and show that a negative response to the PHQ-2 rules out depression efficaciously [45]. The major strength of this study is the relatively large cohort that was recruited over a short period of time, facilitating the investigation of the specific effects of COVID-19-pandemic-related variables.

## 5. Conclusions

Being pregnant may operate as a protective factor against depressive symptoms during a time of crisis, such as the COVID-19 pandemic. Increased social support, engaging in more health behaviors and applying positive appraisal are related to enhanced resilience from depression during pregnancy. Future studies that compare pregnant versus non-pregnant women could clarify the role of pregnancy on women’s mental health during times of hardship. Furthermore, studies would benefit from distinguishing between experiences of depression and anxiety that call for professional help, as opposed to cases in which some degree of sadness, anxiety, fear, anger, and short-term adjustment issues are reasonable and even expected responses [85]. As the pandemic continues, prospective follow-up of women through the perinatal period may yield more information regarding the differential effects of stress during pregnancy and postpartum and should identify the mechanisms of mental health effects of COVID-19. Promoting economic policies and psychosocial interventions to protect pregnant women during the COVID-19 pandemic is essential to ensure positive long-term outcomes and minimize societal and gender gaps. 

## Figures and Tables

**Figure 1 ijerph-18-04298-f001:**
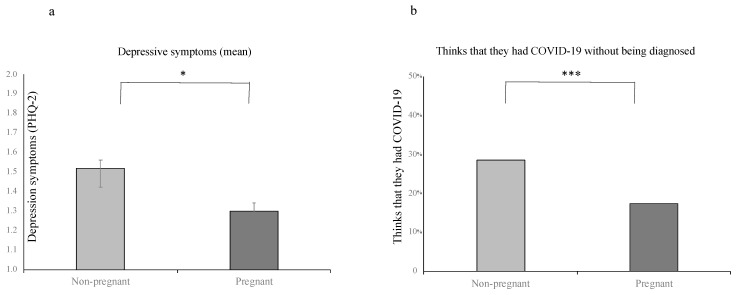
Mental health resiliency in pregnant women. (**a**) Pregnant women show fewer depressive symptoms than non-pregnant women. Depression symptoms were measured using the Patient Health Questionnaire-2 (PHQ-2), which inquires about the frequency of depressed mood and anhedonia over the past two weeks. (**b**) Pregnant women are less concerned that they have COVID-19, even without being diagnosed, compared to non-pregnant women. Note: PHQ-2 = Patient Health Questionnaire-2. * *p* < 0.05, *** *p* < 0.001.

**Table 1 ijerph-18-04298-t001:** Means and standard errors for main study factors among pregnant and non-pregnant women.

Study Variables	Pregnant N = 1114		Non-Pregnant N = 256		χ^2^/*t*-Test (df), *p*	Effect Size
	M/%	S.D.	M/%	S.D.		
**Sociodemographic variables**						
Age	31.88	4.22	35.71	5.42	t_(1366)_ = 12.38, *p* < 0.001	0.79
Years of Education	16.10	2.62	16.59	3.42	t_(1368)_ = 2.15, *p* = 0.03	0.16
Income (Below average)	13.2%		15.6%		X^2^_(1)_ = 1.03, *p* = 0.31	−0.93
Relationship status (Married or cohabiting)	96.5%		85.9%		X^2^_(1)_ = 44.87, *p* < 0.001	0.18
Current employment status (working)	85.7%		85%		X^2^_(1)_ = 0.07, *p* = 0.80	−0.01
Number of children under 18	1.00	1.15	1.93	1.16	t_(1368)_ = 11.60, *p* < 0.001	0.80
**COVID-19 related variables**						
Contact with someone diagnosed COVID-19	8.5%		9.4%		X^2^_(1)_ = 0.18, *p* = 0.67	−0.01
Think they had COVID-19 without being diagnosed	17.4%		28.5%		X^2^_(1)_ = 16.23, *p* < 0.001	−0.11
Access to outdoor spaces (whenever)	84.9%		85.2%		X^2^_(1)_ = 0.01, *p* = 0.91	0.003
**Psychological variables**						
Family/friends support (1–5)	3.93	1.09	3.67	1.15	t_(1368)_ = −3.30, *p* = 0.001	0.23
Partner support (1–5)	4.48	0.78	4.21	0.89	t_(1329)_ = −4.19, *p* < 0.001	0.32
Health behaviors (1–5)	3.27	0.99	2.92	1.10	t_(1368)_ = −5.09, *p* < 0.001	0.33
Stress (1–5)	2.91	1.01	2.86	1.05	t_(1368)_ = −0.64, *p* = 0.52	0.05
Anxiety (GAD-7)	5.96	4.76	5.66	5.65	t_(1368)_ = −0.91, *p* = 0.36	0.06
Depression (PHQ-2)	1.30	1.41	1.52	1.56	t_(1368)_ = 2.06, *p* = 0.04	0.15

**Table 2 ijerph-18-04298-t002:** Multiple hierarchical linear regression analyses—prediction of depression (PHQ-2) score among non-pregnant and pregnant women.

Variables	Step 1 (β)	Step 2 (β)	Step 3 (β)
Step 1: Sociodemographic variables			
Being pregnant	−0.08 *	−0.07 *	−0.06 *
Age	−0.03	−0.02	−0.03
Years of Education	−0.05	−0.05	−0.04
Income (Below average)	0.16 ***	0.13 ***	0.11 **
Relationship status (Married or cohabiting)	0.06	0.06	0.05
Current employment status (working)	0.11 **	0.11 **	0.10 **
Number of children under 18	−0.02	−0.02	−0.09 *
Step 2: COVID-19 related variables			
Contact with someone diagnosed COVID-19		−0.02	−0.01
Think they had COVID-19 without being diagnosed		0.10 **	0.10 **
Access to outdoor space (whenever)		0.06	0.05
Step 3: Social support and health behaviors			
Family/friends support (1–5)			−0.04
Partner support (1–5)			−0.10 **
Health behaviors (1–5)			−0.11 ***
F	7.95 ***	7.16 ***	8.17 ***
R^2^	0.05	0.06	0.09
ΔR^2^	0.05 ***	0.01 **	0.03 ***

* *p* < 0.05, ** *p* < 0.01, *** *p* < 0.001.

**Table 3 ijerph-18-04298-t003:** Multiple hierarchical linear regression analyses—prediction of depression (PHQ-2) score among pregnant women.

Variables	Step 1 (β)	Step 2 (β)	Step 3 (β)	Step 4 (β)	Step 5 (β)
Step 1: Sociodemographic variables					
Age	−0.02	−0.02	−0.03	−0.04	−0.005
Years of Education	−0.05	−0.06	−0.04	−0.03	−0.03
Income (Below average)	0.14 ***	0.13 ***	0.11 **	0.09 *	0.08 *
Relationship status (Married or cohabiting)	0.06	0.06	0.05	0.04	0.03
Current employment status (working)	0.11 **	0.11 **	0.10 **	0.09 *	0.09 **
Number of children under 18	−0.02	−0.01	−0.10 **	−0.10 **	−0.08 *
Step 2: COVID-19 related variables					
Contact with someone diagnosed COVID-19		−0.02	−0.01	−0.01	−0.02
Think they had COVID-19 without being diagnosed		0.09 **	0.08 *	0.08 *	0.06 *
Access to outdoor space (whenever)		0.06	0.04	0.03	0.02
Step 3: Social support and health behaviors					
Family/friends support (1–5)			−0.07	−0.06	−0.05
Partner support (1–5)			−0.09 *	−0.07 *	−0.08 *
Health behaviors (1–5)			−0.14 ***	−0.13 ***	−0.09 *
Step 4: Obstetric factors					
Gestational age (weeks)				−0.06	−0.09 *
Prenatal appointment altered/canceled				0.08 *	0.02
High-risk Pregnancy				0.10 **	0.07 *
Step 5: Pandemic-Related Pregnancy Stress (PREPS)					
Preparedness					0.25 ***
Infection					0.03
Positive Appraisal					−0.07 *
F	7.39 ***	6.26 ***	7.85 ***	7.68 ***	10.11 ***
R^2^	0.05	0.06	0.10	0.12	0.18
ΔR^2^	0.05 ***	0.01 **	0.04 ***	0.03 ***	0.06 ***

* *p* < 0.05, ** *p* < 0.01, *** *p* < 0.001.

## Data Availability

The datasets used and/or analyzed during the current study are available from the corresponding author upon request.

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
