# Peer review of "Women’s Depressive Symptoms during the COVID-19 Pandemic: The Role of Pregnancy"

_ijerph, 2021, doi:10.3390/ijerph18084298_

Round 1

Reviewer 1 Report

Thank you for the invitation to review this manuscript. I really enjoyed reviewing it and the findings are interesting and relevant to our knowledge of how the mental health of reproductive age women (both pregnant and non-pregnant) is impacted by the current pandemic, with important implications for dare I say it, future pandemics.

This is an analysis of an online survey administered to reproductive age women in Israel. It is a well written manuscript. The methods and analyses are appropriate. The authors pose some really interesting research questions and the decision to pursue a comparison of the mental health (and predictors of mental health) in pregnant versus non-pregnant women is novel.

The authors found that non-pregnant women had more mental health symptoms than pregnant women and that one of the predictors of that was fears about COVID-19 infection, which appeared to be higher in non-pregnant group. Some of the risk factors for depression in the pregnant group are factors that we already know increase risk, such as those related to socio-economic deprivation and high-risk pregnancy but some of the predictors such as having fewer children and earlier gestation are relatively less supported by the pre-pandemic literature so they may represent risk factors unique to the pandemic. Likewise, the protective factors of social support and healthy lifestyles are intuitive, but it would be interesting to consider whether or not they exert an even greater protective influence on risk for perinatal mental ill health during a pandemic.

Some questions/minor revisions:

State at the end of your introduction or beginning of your methods that this is a sample in Israel. It is not expressly stated, it is only implied.

The first paragraph of the discussion repeats much of what was already mentioned in the introduction about other studies that have also found reduced risk for anxiety and depression in pregnant women during the pandemic, relative to non-pregnant. It seems unnecessary repetition.

Do you have any further information on ethnicity beyond whether or not the women were Israel-born?

Sentence in limitations: ‘Lastly, PHQ-2 may not be ideal for assessment of depression during pregnancy.’ Can you expand on this? The sentences that follow also need referencing. Also, you focus on the PHQ-2 but it may be more balanced to also consider your other tools e.g. GAD-7 and their relative strengths and limitations.

I would suggest another limitation is the cross-sectional study design; one is really only gaining a snapshot of mental health in May 2020; any symptoms of mental ill health may be transient.

Author Response

We thank you very much for the detailed and helpful comments and appreciate your invitation to revise the manuscript. Please find our point-by-point responses to the reviewers’ comments below. Corrections made in the manuscript in response to the reviewers' comments are marked in red.

Reviewer 1:

  1. State at the end of your introduction or beginning of your methods that this is a sample in Israel. It is not expressly stated, it is only implied.

Response: We have now added to the last paragraph of the Introduction that this is a sample of Israeli women (page 4): “The aim of the current study was to determine the risk for depressive symptoms in pregnant and non-pregnant women in Israel during the COVID-19 pandemic…”.

  1. The first paragraph of the discussion repeats much of what was already mentioned in the introduction about other studies that have also found reduced risk for anxiety and depression in pregnant women during the pandemic, relative to non-pregnant. It seems unnecessary repetition.

Response: We thank the reviewer for this comment. We have now edited the first paragraph of the discussion to avoid redundancy.

  1. Do you have any further information on ethnicity beyond whether or not the women were Israel-born?

Response: Unfortunately, we do not have further information about our participants' ethnicity other than whether or not women were born in Israel and their religion (which were more than 99% Jewish).

  1. Sentence in limitations: ‘Lastly, PHQ-2 may not be ideal for assessment of depression during pregnancy.’ Can you expand on this? The sentences that follow also need referencing. Also, you focus on the PHQ-2 but it may be more balanced to also consider your other tools e.g. GAD-7 and their relative strengths and limitations.

Response: We appreciate the reviewer’s comment and expanded this point in the Limitation section. We now specify that although there are more specific instruments available to evaluate anxiety and depression during the perinatal period (e.g. Postpartum Depression Screening Scale (PDSS) or the Edinburgh Postnatal Depression Scale (EPDS)), this questionnaire was chosen because it applies to both pregnant and non-pregnant women and to not burden participants who were completing additional instruments focused on mood and mental health. We have also added four new references to this limitation (Matthey 2018; Arrieta 2017; Löwe 2005; Kroenke 2003). The Limitation section (page 17) now reads: “Lastly, PHQ-2 may not be ideal for assessment of depression during pregnancy, as there are different instruments that specifically address anxiety and depression during the perinatal period. However, this instrument was chosen as it is a concise and internationally validated screening tool for depression [43], [56], [81], [82]. The psychometric properties are good and show that a negative response to the PHQ-2 rules out depression efficaciously [45]”

Regarding additional tools, we elaborated on the GAD-7 strengths and limitations in the Methods section in the revised manuscript.

  1. I would suggest another limitation is the cross-sectional study design; one is really only gaining a snapshot of mental health in May 2020; any symptoms of mental ill health may be transient.

Response: We added this relevant point to our Limitation section (line 4-7 of the sixth paragraph in the Conclusion) which reads: “Second, it should be noted that this is a cross-sectional study, which provides a snapshot of the mental health of women in May 2020. The results may not necessarily signify a direct causal association between pregnancy and resilience, but rather suggest that other factors that are related to pregnancy…”

Reviewer 2 Report

Despite several concerning study limitations, I did find the topic under investigation to be of certain interest. Please see below for my specific comments.

Specific comments:

  1. "... other studies demonstrate resilience among pregnant and postpartum women during natural disasters [9]" - beyond resilience, there are possible biological or hormonal factors at play. It has often been thought that pregnancy is protective against the development of depression, primarily because of the lower suicide rate during pregnancy and during the 2 years after giving birth (citation: pubmed.ncbi.nlm.nih.gov/14519602). In contrast, the postpartum time period clearly was a period of increased risk for the development of MDD (citation: pubmed.ncbi.nlm.nih.gov/22860768).
  2. In addition to ref [19], suggest authors cite a much more recent systematic review on the effect of yoga-based interventions on depressive symptoms during pregnancy (citation: pubmed.ncbi.nlm.nih.gov/30712750).
  3. What were the inclusion and exclusion criteria for the present study, if any? Were women with high-risk or multiple pregnancies eligible to participate in the study?
  4. How was the questionnaire developed? Was it with reference to international studies on the subject? Did the authors conduct any pilot test for the questionnaire?
  5. Did the authors take steps to guard against duplicate responses to your survey? E.g. IP filtering?
  6. Please provide the actual IRB study/approval number.
  7. Did the authors adjust for baseline depression or anxiety as a covariate? Additionally, socioeconomic status still varies over time in this age range.
  8. What about access to adequate maternal care during a pandemic situation? Would this have played a factor if the health system was more overwhelmed and unable to provide routine maternal care? And what about women with high-risk pregnancies? This and other limitations should be further discussed.
  9. In view of the above, the conclusions should be appropriately tempered. It would be erroneous to assume that being pregnant may reduce the risk of depressive symptoms during a pandemic situation. Women may in fact feel more vulnerable and worry about catching the virus and its lasting effects on a developing fetus. What about social connection? Are pregnant mothers more cared for, with more family members by their side and more attention given to them?
  10. The underlying data should be made publicly available. If this was not possible, please provide a reason why.
  11. A copy of the full questionnaire should be appended in the supplementary material.

Author Response

RE: Women’s Depressive Symptoms During the COVID-19 Pandemic: The Role of Pregnancy

We thank you very much for the detailed and helpful comments and appreciate your invitation to revise the manuscript. Please find our point-by-point responses to the reviewers’ comments below. Corrections made in the manuscript in response to the reviewers' comments are marked in red.

Reviewer 2:

  1. "... other studies demonstrate resilience among pregnant and postpartum women during natural disasters [9]" - beyond resilience, there are possible biological or hormonal factors at play. It has often been thought that pregnancy is protective against the development of depression, primarily because of the lower suicide rate during pregnancy and during the 2 years after giving birth (citation: pubmed.ncbi.nlm.nih.gov/14519602). In contrast, the postpartum time period clearly was a period of increased risk for the development of MDD (citation: pubmed.ncbi.nlm.nih.gov/22860768).

Response: We thank the reviewer for expanding our discussion and contributing relevant literature on this topic. We added to the beginning of the third paragraph of the Introduction two examples of protective biological and hormonal factors during pregnancy. We now cite the suggested manuscript as well: “Research suggests that for some women pregnancy serves as a protective developmental stage, as suggested by the low suicide rate during pregnancy and during the two years after giving birth [23], [24]. Moreover, attenuated emotional and physiological responses to stress during pregnancy may also contribute to enhanced resilience in pregnant women [25], [26]. This may be related to the effect of estrogen on cortisol, as evident by blunted cortisol responses to stress in postmenopausal women receiving estrogen treatment [27], [28].”

  1. In addition to ref [19], suggest authors cite a much more recent systematic review on the effect of yoga-based interventions on depressive symptoms during pregnancy (citation: pubmed.ncbi.nlm.nih.gov/30712750).

Response: Thank you for this interesting suggestion. We now cite the meta-analysis as suggested.

  1. What were the inclusion and exclusion criteria for the present study, if any? Were women with high-risk or multiple pregnancies eligible to participate in the study?

Response: It is now stated in the first paragraph of the Methods section (page 5) that “Inclusion criteria were age 18 years or older and being able to read and write in Hebrew. There were no other exclusion criteria.” Women with high-risk pregnancy and multiple gestation were included, to control for the possibility that these higher-risk situations impacted depression, we controlled for these variables in the regression analysis, and indeed, this was a predictive variable in our study for the development of depressive symptoms during pregnancy.

  1. How was the questionnaire developed? Was it with reference to international studies on the subject? Did the authors conduct any pilot test for the questionnaire?

Response: We now elaborate in the Methods section (page 7) about the questionnaire and state that “the PREPS questionnaire, a novel instrument created by Preis, Brittain, and Lobel [42], a multidisciplinary research and clinical team with expertise in developing validated instruments to assess prenatal maternal stress and coping. Item themes were based on news articles and media interviews regarding women’s experiences during the COVID-19 pandemic, as there was limited research available at the time [42]. Item wording was tested for face validity by pregnant and non-pregnant women before the Stony Brook COVID-19 Pregnancy Experiences (SB-COPE) Study launch. The PREPS questionnaire was translated to several languages, and was found to have good psychometric properties in different populations [46], [47].”

  1. Did the authors take steps to guard against duplicate responses to your survey? E.g. IP filtering?

Response: Indeed. We identified via several ways duplicate responses and chose the first answer in such cases. We now state this in the Methods section (line 8-9 in Participants and Study Design): “After identifying 13 duplicate responses, the final sample included 1,114 pregnant women:”

  1. Please provide the actual IRB study/approval number.

Response: We added to the Institutional Review Board Statement (page 6) the IRB approval number which is 114120.

  1. Did the authors adjust for baseline depression or anxiety as a covariate? Additionally, socioeconomic status still varies over time in this age range.

Response: Thank you for this interesting comment. However, as this is a cross-sectional study, we did not have baseline levels of anxiety and depression, nor socioeconomic status that we could use to control as covariates. We now refer to this in the Limitations section.

  1. What about access to adequate maternal care during a pandemic situation? Would this have played a factor if the health system was more overwhelmed and unable to provide routine maternal care? And what about women with high-risk pregnancies? This and other limitations should be further discussed.

Response: We agree with the reviewer that access to adequate maternal care during a pandemic is a crucial factor in determining mental state. The governmental public health system in Israel was (and still is) stable and well-organized, thus, all medical facilities were open and working full time. However, as many people at the time were worried about coming to the hospitals and outpatient services in-person, many appointments were canceled or delayed. We address this important point in the statistical analysis (fourth stage of our regression analysis now includes “prenatal care appointment canceled or rescheduled owing to the COVID-19 pandemic”, as well as high-risk pregnancies “self-reported high-risk pregnancy status”). To ensure clarity to the readers, we added to the manuscript further discussion on the obstetrical variables in the Conclusion section.

  1. In view of the above, the conclusions should be appropriately tempered. It would be erroneous to assume that being pregnant may reduce the risk of depressive symptoms during a pandemic situation. Women may in fact feel more vulnerable and worry about catching the virus and its lasting effects on a developing fetus. What about social connection? Are pregnant mothers more cared for, with more family members by their side and more attention given to them?

Response: we have now elaborated in the Limitation section on possible factors mediating the relationship between pregnancy and levels of depressive symptoms: “The results may not necessarily signify a direct causal association between pregnancy and resilience, but rather suggest that other factors that are related to pregnancy, such as social connection and support, may mediate the relationship between pregnancy and resilience. Future stages of the current study and other studies will be able to evaluate the course of mental-health symptoms and their sequela throughout the pandemic”.

  1. The underlying data should be made publicly available. If this was not possible, please provide a reason why.

Response: Data will be made available by the authors upon request. While the dataset does not contain personal identifiers (e.g., name, email) it does hold information that can be used to deduce a person’s identity (e.g., number of children, delivery date). Due to the highly sensitive data collected, the risk of confidentiality breach prohibits us from making this data public.

  1. A copy of the full questionnaire should be appended in the supplementary material.

Response:  The Hebrew-version of the questionnaire is now available as Supplementary Materials (Figure 1).

Reviewer 3 Report

Thank you very much for the opportunity to review this manuscript. The manuscript is very interesting and well-constructed. However, the findings of the study are not robust as sample of pregnant and non-pregnant is not comparable. I have the following comments to improve this paper:

  • I urge the authors to address the sample selection bias through Propensity score matching (PSM):
    1. PSM will provide a comparable sample on which authors can make comparison.
    2. Authors rerun the analysis by eliminating the non-matched observation.
  • Authors add demographic variables as control variable in each regression model as R2 of regression models is very low.
  • Findings of the study may be different after PSM which may leads to significant changes in the conclusion and findings of the study.
  • Authors cite more recent studies in the introduction and discussion from the year 2021.
  • Conclusion section is short. I suggest that authors add implications and future research directions at the end of the conclusions section.
  • Finally, the authors check the minor issues and typos.

Author Response

RE: Women’s Depressive Symptoms During the COVID-19 Pandemic: The Role of Pregnancy

We thank you very much for the detailed and helpful comments and appreciate your invitation to revise the manuscript. Please find our point-by-point responses to the reviewers’ comments below. Corrections made in the manuscript in response to the reviewers' comments are marked in red.

  1. I urge the authors to address the sample selection bias through Propensity score matching (PSM):
    1. PSM will provide a comparable sample on which authors can make comparison.
    2. Authors rerun the analysis by eliminating the non-matched observation.

Authors add demographic variables as control variable in each regression model as Rof regression models is very low.

Findings of the study may be different after PSM which may leads to significant changes in the conclusion and findings of the study.

Response: Following the Reviewer's valuable suggestion, we have complemented the "traditional" regression analyses with a regression based on propensity-matched groups using demographic variables as covariates. As can be seen in Supplementary Figures 2-3 and Tables 3-4, limiting the regression to optimally-matched groups yielded a highly significant effect of pregnancy on PHQ2. As the effects were similar to those observed using non-matched samples, we conclude that the effects of pregnancy seem unlikely to be principally driven by demographic differences between groups.

  1. Authors cite more recent studies in the introduction and discussion from the year 2021.

Response: We have now added more recent studies in the Introduction and Discussion sections from the end of 2020 and 2021, including: Pierce et al., 2020; Racine et al., 2021; Perzow et al., 2021; Thompson et al., 2021; Lubián López et al., 2021; Preis et al., 2021; Brik et al., 2021; Niela-Vilén et al., 2021 and others.

  1. Conclusion section is short. I suggest that authors add implications and future research directions at the end of the conclusions section.

Response: Thank you for this comment. We elaborated on the implications and future research as well as on other topics in the Conclusion section, including the mediating effect of social support and governmental policies in the limitation paragraph and our findings regarding obstetrical variables in the fourth paragraph. At the end of the manuscript, we added that “as the pandemic continues, prospective follow-up of women through the perinatal period may yield more information regarding the differential effects of stress during pregnancy and postpartum.  and should identify the mechanisms of mental health effects of COVID-19.”

  1. Finally, the authors check the minor issues and typos.

Response: We proofread the article to eliminate any typos and other minor issues.  

Round 2

Reviewer 2 Report

Thank you for the revisions.

Specific comments:

  1. Some background about the narrative of COVID-19 in Israel, e.g. number and distribution of COVID-19 cases may be helpful to contextualize the present study.
  2. The authors should include a "discussion" section rather than lump everything under "conclusions". The conclusions should be more succinctly stated. Most of the paragraphs under the conclusion section should be moved to a new discussion section.
  3. Rather than over-pathologize these experiences, some degree of sadness, anxiety, fear, anger, paranoia, and short-term adjustment issues and long-term adaptation to the uncertain future are perhaps reasonable or expected responses. The majority of mental disorders following COVID-19 may be “reactive” in nature. It may be in response to the fear and stress of contagion, especially given the possibility of asymptomatic spreaders in the community. It may be the consequence of hospitalization for infected individuals/family members or strict measures to curb and contain the pandemic, with “lockdown” living, loss of livelihood, and financial hardship (citation: pubmed.ncbi.nlm.nih.gov/32943541). It is clear that this pandemic has disproportionately impacted racial minorities and lower-income families (citation: pubmed.ncbi.nlm.nih.gov/32391864).
  4. "... pandemic had a severe impact on mental health and wellbeing of the general population" - suggest to omit/replace the word "severe".
  5. Please change "A Finish study" to "A Finnish study".

Author Response

1. Some background about the narrative of COVID-19 in Israel, e.g. number and distribution of COVID-19 cases may be helpful to contextualize the present study.

Response: The first sentences of the Discussion give background regarding the atmosphere in Israel at the time of the study. Furthermore. we have added to the Limitation section more specific information regarding COVID-19 in Israel during the time the research was conducted: “During that time, there was a relatively small number of COVID-19-related infections and death cases (an average of 2.188 new infections and 0.233 new death cases per million people per day [81]). However, the lockdown and movement restrictions applied as well as the uncertainty regarding the influence of the COVID-19 pandemic worldwide created an atmosphere of stress and anxiety.”.

2. The authors should include a "discussion" section rather than lump everything under "conclusions". The conclusions should be more succinctly stated. Most of the paragraphs under the conclusion section should be moved to a new discussion section.

Response: we have now changed the headline “Conclusions” to “Discussion” and only the last paragraph is now under the title “Conclusions”.

3. Rather than over-pathologize these experiences, some degree of sadness, anxiety, fear, anger, paranoia, and short-term adjustment issues and long-term adaptation to the uncertain future are perhaps reasonable or expected responses. The majority of mental disorders following COVID-19 may be “reactive” in nature. It may be in response to the fear and stress of contagion, especially given the possibility of asymptomatic spreaders in the community. It may be the consequence of hospitalization for infected individuals/family members or strict measures to curb and contain the pandemic, with “lockdown” living, loss of livelihood, and financial hardship (citation: pubmed.ncbi.nlm.nih.gov/32943541). It is clear that this pandemic has disproportionately impacted racial minorities and lower-income families (citation: pubmed.ncbi.nlm.nih.gov/32391864).

Response: we have added the point raised by the reviewer and added to the Conclusion section as a future direction and state that “Furthermore, studies would benefit from distinguishing between experiences of depression and anxiety that call for professional help, as opposed to cases in which some degree of sadness, anxiety, fear, anger, and short-term adjustment issues are reasonable and even expected responses.  [84].”  We also added to the Limitations Section the reviewer’s suggestion regarding the substantial impact of the pandemic on racial minorities and lower-income families, which are underrepresented in our cohort

4. "... pandemic had a severe impact on mental health and wellbeing of the general population" - suggest to omit/replace the word "severe".

Response: we have replaced the word “severe” to “notable”.

5. Please change "A Finish study" to "A Finnish study".

Response: done.

Reviewer 3 Report

Accept in present form.

Author Response

Thank you very much